# Aspirin: A Suicide Inhibitor of Carbonic Anhydrase II

**DOI:** 10.3390/biom10040527

**Published:** 2020-03-31

**Authors:** Jacob Andring, Jacob Combs, Robert McKenna

**Affiliations:** Department of Biochemistry and Molecular Biology, College of Medicine, University of Florida, Gainesville, FL 32610, USA; jacobandring@ufl.edu (J.A.); jacob.combs@ufl.edu (J.C.)

**Keywords:** carbonic anhydrase, Aspirin, salicylic acid, esterase, X-ray crystallography

## Abstract

Carbonic anhydrase II (CAII) is a metalloenzyme that catalyzes the reversible hydration/dehydration of CO_2_/HCO_3_^−^. In addition, CAII is attributed to other catalytic reactions, including esterase activity. Aspirin (acetyl-salicylic acid), an everyday over-the-counter drug, has both ester and carboxylic acid moieties. Recently, compounds with a carboxylic acid group have been shown to inhibit CAII. Hence, we hypothesized that Aspirin could act as a substrate for esterase activity, and the product salicylic acid (SA), an inhibitor of CAII. Here, we present the crystal structure of CAII in complex with SA, a product of CAII crystals pre-soaked with Aspirin, to 1.35Å resolution. In addition, we provide kinetic data to support the observation that CAII converts Aspirin to its deacetylated form, SA. This data may also explain the short half-life of Aspirin, with CAII so abundant in blood, and that Aspirin could act as a suicide inhibitor of CAII.

## 1. Introduction

Carbonic anhydrases (CAs) are a family of mainly zinc metalloenzymes responsible for the interconversion of carbon dioxide (CO_2_) into bicarbonate (HCO_3_^−^) and a proton via a ping-pong mechanism [Equation (1)] [1]. As such, CAs play an important role in blood homeostasis, CO_2_/HCO_3_^−^ transportation, and pH regulation [2]. There are 12 catalytic isoforms of CA expressed in humans, each with unique amino acid sequences, catalytic rates, cellular location, and tissue expression [2]. The active site of human CAs is conserved, with a zinc ion coordinated by three histidine (H94, H96, and H119 (CAII numbering)) and a water/hydroxide [3]. Of these isoforms, CAII is the most widely expressed isoform, responsible for regulating the intracellular pH in nearly every cell [4]. CAII is the fastest human CA, with a kcat of ~1100ms^−1^ that approaches the rate of diffusion [5].

CAs play a critical role in physiology, to increase the rate of CO_2_/HCO_3_^−^ interconversion (Equation (1)) [4]. HCO_3_^−^ is the most commonly transported form of CO_2_ in the body [4]. Large quantities of CO_2_ are produced in tissues during respiration before removal by red blood cells (RBC) and transported to the lungs [4]. While CAII plays a large role in transporting CO_2_, it isn’t the only mode of excretion. CAII expression levels are elevated in the kidney as it regulates HCO_3_^−^ flux [6]. CAII also balances cytoplasmic pH via interactions with a variety of membrane-bound ion carriers, including MCT1 and 4 [4].

In addition, CAII is important in blood homeostasis [4]. Human RBCs contain a high concentration of CAII at 0.8 attomol [7]. CAII has also been shown to be involved in regulating platelet function. While the exact mechanism is unknown, CAII is known to be involved in nitrocysteine and nitric oxide formation, both critical in platelet inhibition [8].

As CAs are responsible for a variety of physiological functions and pH regulation, they are often clinically targeted. CA inhibitors (CAIs) are used to treat a variety of diseases such as glaucoma, altitude sickness, and epilepsy [9]. In addition, CAIs are currently being developed as anti-cancer drugs [10,11,12]. These inhibitors are designed to bind to the active site zinc, displacing the zinc bound solvent. The most common type of CAIs are sulfonamides, such as acetazolamide, which has nM binding affinity. Many of these sulfonamide-based molecules are used clinically, such as dorzolamide, for the treatment of glaucoma [13,14]. In addition to sulfonamides, a variety of other chemical motifs have been identified to inhibit CA, such as carboxylic acids [15]. Nicotinic and ferulic acid have recently been identified as inhibitors of CAII [16]. Unlike the sulfonamide-based drugs, these inhibitors do not directly displace the zinc bound solvent, but instead anchor through the solvent, blocking substrate entry to the active site [16]. Furthermore, 3-nitrobenzoic acid has also been reported as a potent CAI, with further studies showing its potential clinical relevance as a cancer therapeutic [17]. Previous work has also shown that salicylic acid as well as some phenol derivatives are μM inhibitors of mammalian CAs, although the exact mechanism of inhibition for salicylic acid is unknown [18]. These carboxylic acid-based compounds represent a new and largely unstudied class of CAIs.

Aspirin (acetylsalicylic acid) is one of the most widely studied and consumed drugs in use. Aspirin is a known cyclooxygenase (COX) inhibitor, giving the molecule its anti-inflammatory and blood thinning characteristics [19]. Aspirin inhibits the COX enzymes by acetylating critical active site residues, leaving the enzymes acatalytic while generating salicylic acid (SA) as a byproduct [19]. While Aspirin is typically used by patients prone to heart disease, there are many hypotheses about its other potential therapeutic benefits, such as a chemotherapy or a preventive of preeclampsia [19,20,21]. Each year, 4 × 10^4^ metric tons of Aspirin are consumed, which equate to ~120 billion pills [21]. A typical dose of Aspirin is 325 mg; however, there are lower dosage options for everyday use and higher concentrations (up to 6 g per day, or ~7 mM in blood) for at-risk patients with heart disease [22]. Interestingly, Aspirin only has a half-life of ~15 min in blood due to a previously unidentified carboxylesterase [23]. The short half-life of Aspirin leads to patients taking the drug daily to keep a therapeutic dose in their system. A recent study found in a genome-wide search that CAII is the only protein overexpressed in patients with Aspirin resistance and therefore may be the unidentified carboxylesterase [24]. Since Aspirin is a carboxylic acid-based molecule, it was hypothesized that it could potentially bind to CAII. Here, we examine this hypothesis through structure activity relationship studies between Aspirin and CAII, through X-ray crystallography and a spectroscopy-based kinetic assay. We determine that CAII is the previously unidentified carboxylesterase responsible for Aspirin’s short half-life in the blood, and that the product of this reaction, SA, can then inhibit CAII, thus making Aspirin a suicide inhibitor.
(1)$$\mathrm{Aspirin}+H_{2}O\;\overset{\mathrm{CAII}}{\Rightarrow }\;\mathrm{SA}\cdot \mathrm{CAII}+\mathrm{Acetate}$$

## 2. Materials and Methods

### 2.1. Protein Expression and Purification

CAII was expressed and purified according to previously published protocols [25,26,27]. Competent BL21(DE3) cells were transformed with 1:l plasmid DNA containing the *CA2* gene under expression control of T7 promoter. Cells were heat shocked for 45 s at 42 °C, then placed on ice for two minutes; 350 μL of Luria broth (LB) was added and grown at 37 °C at 200 rpm for one hour. An overnight culture at 37 °C was used the following day for large-scale growth until the OD600 reached ~0.6. Protein expression was induced by the addition of 1 mL of 100mg/mL Isopropyl-β-d-thiogalactopyranoside (IPTG) for three hours; 1 mL of 1M zinc sulfate was also added to aid in the folding of CAII. The culture was centrifuged for 10 min at 5000 rpm and the pellets were frozen overnight. The pellets were thawed and suspended with 40 mL of Wash Buffer 1 (WB1, 0.2 M sodium sulfate, 0.1 M Tris–HCl, pH 9.0). Then, 40 mg of lysozyme and 5 mg of DNaseI were added into the bottle, then stirred at 4 °C for one hour. A microfluidizer (LM10, Microfluidics, Boston, MA, USA) lysed the cells before centrifugation at 1.2 × 10^4^ rpm for one hour at 4 °C, and then the supernatant was filtered with a 0.45-μm filter (09-720-4, Fisherbrand, Boston, MA, USA).

Subsequently, p-Aminomethylbenzenesulfonamide agarose resin affinity column was set up and equilibrated with WB1. The lysate was then loaded onto the column and washed with WB1 and Wash Buffer 2 (0.2 M sodium sulfate, 0.1 M Tris–HCl, pH 7.0) to elute non-specific proteins. The CA was eluted off the column with 0.4M sodium azide in 50 mM Tris–HCl, at pH 7.8. Eluent was added to Amicon (Ultra-15 centrifugal filter, Millipore Sigma, Darmstadt, Germany) devices with a 10^4^ -kDa molecular weight cutoff and centrifuged at 6,000 rpm for 15 min, reducing the volume to ~2 mL. Then, 10 mL of storage buffer (50 mM Tris–HCl, pH 7.8, filtered) was added, spun at 6 × 10^3^ rpm for 15 min, and the solution was resuspended. This was repeated five times to fully remove the azide. Final protein concentration was checked by measuring the absorbance at 280 nm. A 12% SDS-PAGE gel was prepared to analyze protein purity.

### 2.2. X-ray Crystallography

#### 2.2.1. Crystallization

Prior to crystallization, purified CAII was concentrated to 10 mg/mL via Amicon Ultra-15 centrifugal filters. CAII was crystallized via the hanging drop vapor diffusion method; 2.5 μL of 10 mg/mL protein was added to siliconized glass cover slips along with 2.5 μL of mother liquor consisting of 1.6 M sodium citrate and 50 mM Tris at pH 7.8; 500 μL of mother liquor was added to the wells and grease was used to seal the glass clover slips to the wells [28]. Crystals formed within 24 h. The 500 mM Aspirin was purchased through Sigma Aldrich (St. Louis, MI, USA) and salicylic acid was purchased through Fisher Scientific (Lenexa, KA, USA). Each chemical was determined to be >99% purity through NMR and other assays. Aspirin was dissolved in 100% ethanol and a 1:10 dilution was made for a final concentration of 50 mM Aspirin in 10% ethanol; 1 μL of the Aspirin solution was added to the CAII drops and allowed to soak for 20 min.

#### 2.2.2. Data Collection

The soaked Aspirin CAII crystals were harvested, flash frozen in liquid nitrogen, and shipped to Stanford Synchrotron Radiation Lightsource (SSRL). Data was collected at the 9-2 beamline at SSRL, using a Pilatus 6M detector (Dectris, Philadelphia, PE, USA) with 0.15° oscillations, a wavelength of 0.9795Å, and a detector distance of 250 mm. Each data set consisted of 1200 images for a total of 180° data.

#### 2.2.3. Data Processing

The diffraction images were indexed and integrated using XDS (Heidelberg, Germany), then merged and scaled to the P21space group, using the program Aimless via the CCP4 program suite (7.70.076, UK) [29,30,31]. The diffraction data was phased using the software package PHENIX (1.14, Berkeley, CA, USA) utilizing the high resolution CAII PDB entry 3KS3 as the search model [32]. Coordinate refinements were calculated using PHENIX, while the program Coot (0.8.9.2, York, England) was utilized to add solvent and SA [32,33]. Coot was also utilized to make individual real space refinements of each residue where appropriate [33]. Aspirin modeling into hCAII was done in Chimera (1.11.2, San Francisco, SA, USA) with MMTK providing minimization routines [34]. Performing an energy minimization for Aspirin allows for a more accurate depiction for how the drug may bind in the active site. Protein–inhibitor interactions were determined using LigPlot Plus (2.2, Cambridge, England) and figures were made in the molecular graphical software PyMol (2.0.5, Cambridge, England) [35,36].

### 2.3. CA Inhibition Studies

Esterase assays were performed to measure the inhibition constants of Aspirin and SA on CAII using 4-nitrophenyl acetate as a colorimetric substrate. CAII cleaves the ester bond of 4-nitrophenyl acetate, generating 4-nitrophenol. The product, 4-nitrophenol, absorbs strongly at 348 nm, thus the reaction can be monitored spectroscopically [37]. CAII has high levels of esterase activity due to the nucleophilic nature of the zinc bound hydroxyl.

In a 96 deep-well plate, 50 μL of 0.1 mg/mL CAII (concentration in the 50 μL-sample well) in storage buffer was added to each well. For inhibition studies, varying concentrations of inhibitors were preincubated with CAII at room temperature for 20 min prior to testing. To initiate the reaction, 200 μL of 0.8 mM pNPA dissolved in 3% acetone in water was added to the sample well. The well plate was then immediately inserted into the plate reader (Synergy HTX, BioTek, Winooski, WI, USA). Absorbance at 348 nm was recorded every 8 s for 10 min; 100 nM and 1000 nM acetazolamide were used as a positive control for inhibition. Inhibition with 100 nM acetazolamide showed 45% activity while 1000 nM showed 3.1% activity.

## 3. Results

Based on previous studies with CAIs and the knowledge that CAII may be involved with Aspirin resistance, X-ray crystallography was utilized to determine if Aspirin can bind in the CAII active site. Suitable CAII crystals were grown using the sitting–drop, vapor diffusion method, and soaked with a solution of 50 mM Aspirin. The CAII crystals were well ordered and diffracted to 1.35Å resolution. Upon data analysis and refinement, surprisingly, SA was observed bound to the CAII active site instead of the expected Aspirin (Figure 1). These data have been submitted to the Protein Data Bank with PDB code 6UX1 and the crystallization statistics are reported (Appendix A). While the benzene ring and carboxylic acid group showed clear electron density, the ester linked acetate group was absent. Like the previously determined carboxylic acid based inhibitor complexes, SA was shown to bind through the zinc bound solvent, and not the displacement of it. The carboxylic acid motif binds to the zinc bound solvent in the same orientation as the substrate CO_2_ [38]. SA is stabilized within the active site through interactions with residues on both the hydrophobic face and the hydrophilic face. On the hydrophilic face, the gatekeeper residue T199 as well as T200 form three hydrogen bonds with the carboxylic acid of SA. As well, Q92 forms distant dipole–dipole interactions with the hydroxyl of SA. On the hydrophobic face, residues V121, F131, and L198 form multiple Van der Waals interactions with the ring of SA. In addition, several other SA molecules were bound in various pockets on the surface of CAII; however, these were involved in crystal lattice packing interactions and therefore not further discussed.

A known and widely studied function of CAs is their ability to act as an esterase [37]. When a small molecule with an ester bond such as Aspirin enters the active site of CAII, its ester bond is cleaved leaving an acetyl group and SA in the case of Aspirin. The zinc bound hydroxide is a strong nucleophile, able to attack the carbonyl of an ester, cleaving the bond. This esterase activity is often used to measure the activity of CAs, by monitoring the reaction via a colorimetric probe. The compound 4-nitrophenyl acetate or pNPA is often used to monitor this reaction as its ester bond is cleavable by CAII and its product, 4-nitrophenol, is spectroscopically absorbent at 348 nm. Therefore, this molecule is used as a substrate of CAII and its cleavage is monitored by measuring absorbance at 348 nm. Inhibitors can then be added to determine their efficacy at inhibiting CAII [39,40].

Kinetic experiments were performed for both Aspirin and SA individually in the presence of CAII. Based on the crystallography experiments, it was predicted that Aspirin would act as a substrate firstly, then form SA, thus acting as an inhibitor. The results from our preliminary kinetic assays with Aspirin, however, were inconclusive. The difference in absorption spectra between Aspirin and SA was problematic as the molecules absorbed differently at the experimental wavelength of 348 nm. Thus, the background absorbance of Aspirin and SA convoluted the absorbance from the esterase substrate, pNPA. Therefore, SA was tested by itself to circumvent the background absorbance from the Aspirin cleavage. At physiologically relevant concentrations of SA, it was shown that SA completely inhibited CAII (Figure 2). Using prism 8 software, the data was plotted to a non-linear regression to determine IC50 [41]. The average standard deviation was 2.2% and the standard deviation specifically at 50% activity is 1.5%. With the small deviation in percent activity, we can confidently say that the IC50 is 6.6 +/− 0.5mM (Figure 2). Therefore, we show that at a clinically prescribed high dosage of Aspirin, CAII can act as an Aspirin esterase to form SA, which can then act as a suicide inhibitor.

Using molecular modeling, it can be hypothesized that Aspirin would bind in a similar fashion to nicotinic and ferulic acid within the active site of CAII, priming its ester group for nucleophilic attack (Figure 3) [16]. The fast rate of conversion from Aspirin to SA has made it difficult to obtain the crystal structure of the Aspirin–CAII complex [16].

Similarly to SA, Aspirin is predicted to bind through the zinc bound solvent. The acetyl portion was bound to the zinc bound solvent based on the previously solved structure of acetate binding to CAII [42]. The carboxylic acid motif, however, is positioned towards the hydrophilic pocket, interacting with T199 and T200. In the hydrophobic face, residues V121, V143, L198, and W209 form multiple Van der Waals interactions with the ring of Aspirin. The interaction with Q92 is conserved; however, F131 is positioned too far for interactions.

## 4. Discussion

Based on the crystallographic data of SA bound to CAII and the modeling with Aspirin, we propose a mechanism for CAII Aspirin ester cleavage and SA inhibition (Figure 4). Firstly, Aspirin binds to the zinc bound solvent within the active site with its acetate bound in a similar fashion of CO_2_ binding, positioned for nucleophilic attack (Figure 4A). The hydroxyl cleaves the ester bond in the Aspirin molecule leaving the acetate bound to the active site (Figure 4B). The acetate of the reaction is displaced by a water molecule that binds the zinc, while the SA remains in the active site (Figure 4C,D). Finally, the SA reorients within the active site and anchors through the zinc bound solvent, inhibiting any further reaction (Figure 4E). This mechanism would explain how Aspirin initially acts as a substrate for CAII esterase activity, then its product, SA, is able to inhibit the enzyme.

## 5. Conclusions

Based on these findings, we conclude that Aspirin binds and inhibits CAII via the SA product, as it retains the carboxylic acid motif similar to other CAIs such as nicotinic, ferulic, and 3-nitrobenzoic acids. These findings imply CAII’s importance in the blood, beyond its carbonic anhydrase activity and further implicate the enzyme in platelet function. We have identified CAII as the carboxylesterase responsible for Aspirin’s short half-life in the blood. Therefore, perhaps a combined therapy with a CA inhibitor and Aspirin could improve Aspirin’s efficacy in the treatment of heart disease.

## Figures and Tables

**Figure 1 biomolecules-10-00527-f001:**
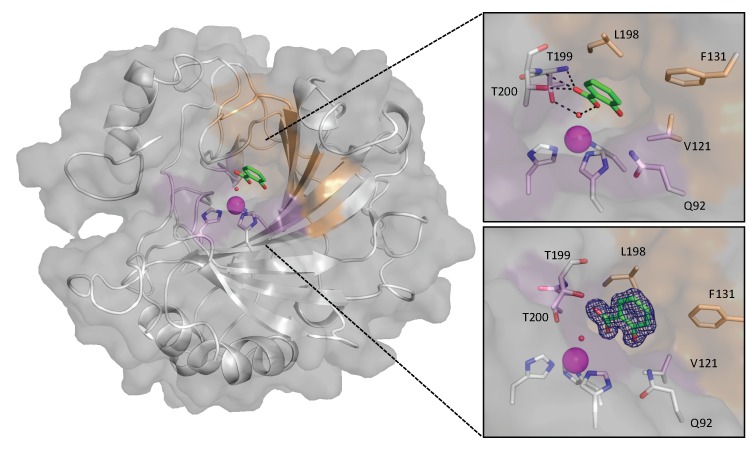
Structure of carbonic anhydrase II (CAII) complexed with salicylic acid (SA). The hydrophobic face of CAII is shown as the orange surface while the hydrophilic face is shown as violet. Zinc is depicted as a magenta sphere with critical binding residues shown in sticks. Bound salicylic acid is shown in green sticks. Top insert, active site with SA interactions and hydrogen bonds shown in dashes; bottom insert, electron density for SA shown as blue mesh. PDB: 6UX1.

**Figure 2 biomolecules-10-00527-f002:**
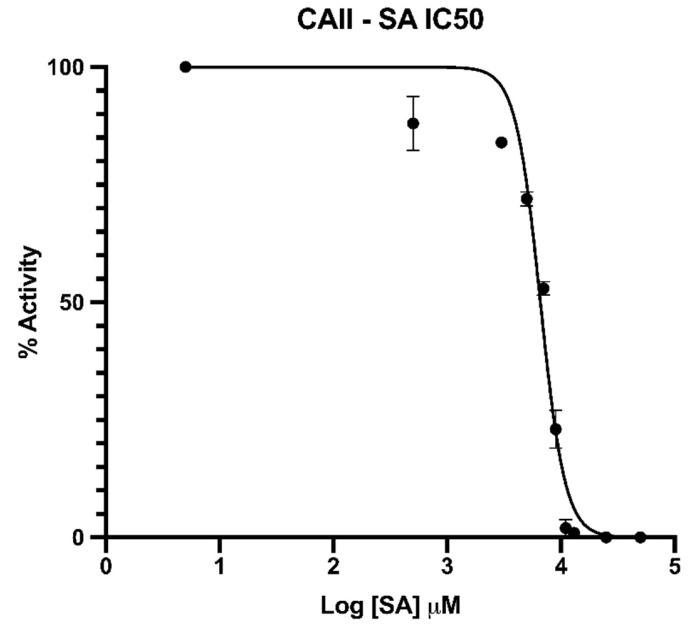
Inhibition curve of CAII with SA. Calculated IC50 of 6.6 mM. The error bars represent the standard deviation of 3 kinetic experiments performed.

**Figure 3 biomolecules-10-00527-f003:**
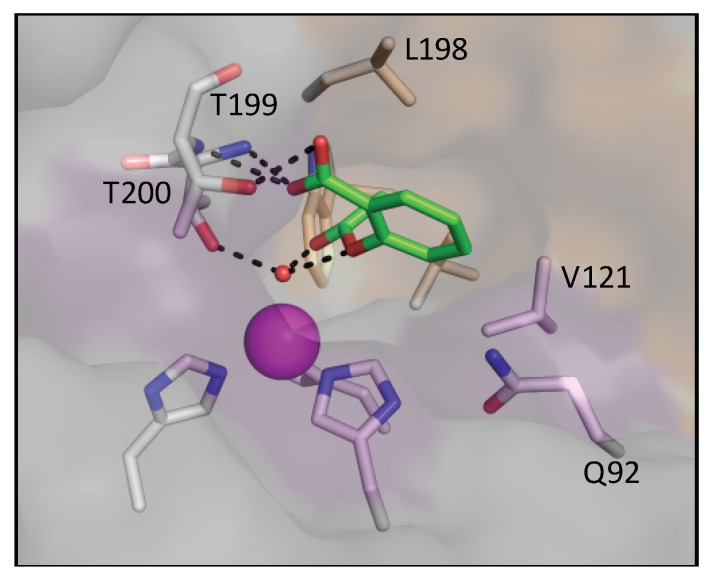
Structure of Aspirin modeled into the active site of CAII. The hydrophobic face of CAII is shown as orange surface while the hydrophilic face is shown as violet. Zinc is depicted as a magenta sphere with critical residues shown in sticks. Bound Aspirin is shown in pink sticks. V134 and W204 are unlabeled for clarity.

**Figure 4 biomolecules-10-00527-f004:**
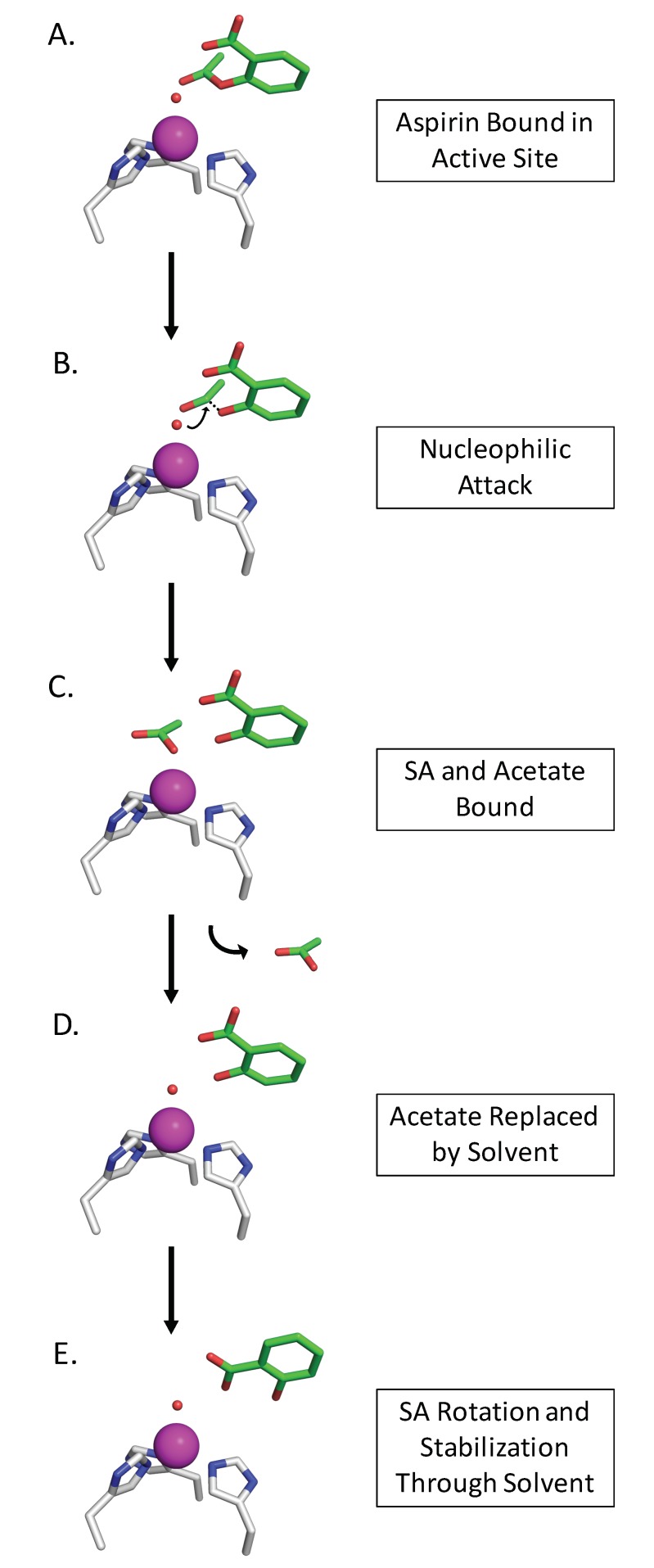
Proposed mechanism of CA esterase function, which converts Aspirin to SA. (**A**) Aspirin binds through the zinc bound hydroxyl orienting its acetyl group to how acetate binds in CAII (PDB: 1CAY). (**B**) The ester bond in Aspirin is cleaved via nucleophilic attack. (**C**) Once the ester bond is cleaved, acetate is generated and briefly is bound to the zinc. (**D**) A solvent molecule replaces acetate on the zinc and acetate is released to bulk solvent. (**E**) Salicylic acid rotates to bind its carboxylic acid moiety to the zinc bound solvent, stabilizing it within the active site.

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
