# Peer review of "Aspirin: A Suicide Inhibitor of Carbonic Anhydrase II"

_biomolecules, 2020, doi:10.3390/biom10040527_

Round 1

Reviewer 1 Report

The manucript from Andring et al. is an interesting contribution in the field of carbonic anhydrase. In the manuscript, a crystal structure of the complex between CA II and salicylic acid is demonstrated for the first time. This paper helps us to understand how CA may participate in the conversion from Aspirin to salicylic acid. The manuscript generally reads well and the findings are of interest. My main comment is that the authors should carefully read and cite the paper from Innocenti et al. "Carbonic Anhydrase Inhibitors: Inhibition of Mammalian Isoforms I-XIV With a Series of Substituted Phenols Including Paracetamol and Salicylic Acid." This paper from 2008 demonstrated that salicylic acid is an efficient inhibitor of most human CA isoenzymes. Innocenti et al. already proposed a mechanism for binding. The authors could now discuss how their findings correlate to the previous findings. 

Minor comments:

The text has many extra hyphens: e.g. de-acetylated, trans-ported, proto-cols, ul-tra, pro-gram, per-forming, ab-sorbs, inter-actions, with-in, sol-vent

3-nitro benzoic ... should be 3-nitrobenzoic

Line 66: "A recent study found in a genome wide search found that..."

Lines 70-71: extra paragraph is not needed here

Line 78: CAII gene ... should be CA2 gene (in Italics)

Line 97: "protein purification purity" does not sound OK

Line 141-142: refers to previous studies... a reference should be added

Line 221: CAIIs ... should be CAII´s 

Lines 223,224: Aspirins... should be Aspirin´s 

Reviewer 2 Report

This is a very good paper proving that aspirin is acting on carbonic anhydrase II as a suicide inhibitor (or suicide substrate) because it is firstly hydrolyzed and then exerts inhibitory activity. Paper is sound and well written, however, requires more careful editing.

Below are the editorial erros I had found, in order of their appearance in manuscript:

1./ merge text in lines 70 and 71;

2./ protocols not proto-colas (line 77);

3./ p-Aminomethyl... not P-aminomethyl... (line 89);

4./ absorbs not ab-sorbs (line 129);

5./ CO2 notCO2 (line 151);

6./ Nicotinic and ferulic acids have to be writtem in small letters (all text);

7./ within not with-in (line 192);

8./ Aspirin's not Aspirins (lines 223 & 224);

9./check reference 8 (line 264).
